# Intraoperative 3D Imaging Reduces Pedicle Screw Related Complications and Reoperations in Adolescents Undergoing Posterior Spinal Fusion for Idiopathic Scoliosis: A Retrospective Study

**DOI:** 10.3390/children9081129

**Published:** 2022-07-28

**Authors:** Antti J. Saarinen, Eetu N. Suominen, Linda Helenius, Johanna Syvänen, Arimatias Raitio, Ilkka Helenius

**Affiliations:** 1Department of Pediatric Orthopedic Surgery, Turku University Hospital, 20521 Turku, Finland; anjusaa@utu.fi (A.J.S.); ensuom@utu.fi (E.N.S.); johanna.syvanen@tyks.fi (J.S.); arimatias.raitio@fimnet.fi (A.R.); 2Department of Orthopedics and Traumatology, Helsinki University Hospital, 00260 Helsinki, Finland; 3Department of Anesthesia and Intensive Care, Turku University Hospital, 20521 Turku, Finland; linda.helenius@tyks.fi

**Keywords:** scoliosis, O-arm, navigation, pedicle screws, complication, spinal fusion, adolescent idiopathic scoliosis

## Abstract

Widely used surgical treatment for adolescent idiopathic scoliosis (AIS) is posterior spinal fusion using pedicle screw instrumentation (PSI). Two-dimensional (2D) or three-dimensional (3D) navigation is used to track the screw positioning during surgery. In this study, we evaluated the screw misplacement, complications, and need for reoperations of intraoperative 3D as compared to 2D imaging in AIS patients. There were 198 adolescents, of which 101 (51%) were evaluated with 2D imaging and 97 (49%) with 3D imaging. Outcome parameters included radiographic correction, health-related quality of life (HRQOL), complications, and reoperations. The mean age was 15.5 (SD 2.1) years at the time of the surgery. Forty-four (45%) patients in the 3D group and 13 (13%) patients in the 2D group had at least one pedicle screw repositioned in the index operation (*p* < 0.001). Six (6%) patients in the 2D group, and none in the 3D group had a neurological complication (*p* = 0.015). Five (5%) patients in the 2D group and none in the 3D group required reoperation (*p* = 0.009). There were no significant differences in HRQOL score at two-year follow-up between the groups. In conclusion, intraoperative 3D imaging reduced pedicle screw-related complications and reoperations in AIS patients undergoing PSI as compared with 2D imaging.

## 1. Introduction

Scoliosis is a spinal deformity, which refers to the deviation of the spine greater than 10° in the coronal plane. The most common type of scoliosis is idiopathic scoliosis, which, based on the age of onset, can be classified as infantile, juvenile, and adolescent. Adolescent idiopathic scoliosis (AIS) is the most common form of idiopathic scoliosis, affecting 1–3% of children in the at-risk population of those aged 10–16 years [1,2]. Although most adolescents with AIS will not develop pain or neurologic symptoms, scoliosis can progress to rib deformity and respiratory compromise and can cause significant cosmetic problems and emotional distress. Treatment options for AIS include observation, bracing, and surgery. The goals of treatment for pediatric idiopathic scoliosis are to correct deformity, prevent curve progression, and restore trunk symmetry and balance [1,2].

Posterior corrective surgeries using pedicle screw instrumentation (PSI), which allows curves to be corrected in three dimensions, have become the most popular surgical treatment for scoliosis [3,4]. Pedicle screws provide better deformity correction, improved pulmonary function, reduced blood loss, shorter fusion levels, and reduced operation time compared to prior correction methods [4,5,6,7,8]. Posterior spinal fusion (PSF) is carried out using subperiostal exposure, bilateral facetectomy, and application of bone graft on decorticated posterior spinal bony elements. To address scoliosis the spine is instrumented bilaterally using pedicle screws and these are connected with rods, which allows the straightening of the spine and preventing the curve from progression. PSF with pedicle screws is the gold-standard treatment for AIS. It allows three-dimensional correction of the spinal deformity, and furthermore, with pedicle screw instrumentation provides good correction of the scoliosis with low reoperation rates [3,4]. There is evidence, that patients who underwent posterior spinal fusion with pedicle screws experienced improved back pain and health-related quality of life compared with patients with untreated AIS [5]. In contrast to the comparative efficacy of this technique in correcting scoliosis, screw placement has increased risks of causing complications, such as neurological or vascular injuries [6,7].

High screw density helps restore thoracic kyphosis and correct spinal rotation [8]. Computer tomography (CT)—based navigation methods have improved results on the radiographic outcomes and reduced the number of misplaced screws significantly [9,10]. C-arm fluoroscopic imaging has been used for years and has enabled relatively accurate placement of pedicle screws. However, the C-arm system only provides two-dimensional (2D) fluoroscopic images, for which additional artificial correction during the operation is required in cases with severe deformities due to scoliosis. The flexibility of the growing spine and anatomical deformities caused by AIS reduce navigation accuracy, and even navigated screws are sometimes misplaced [11]. The use of modern three-dimensional (3D) intraoperative imaging and navigation techniques have been introduced for the accurate insertion of pedicle screws (PS) for the growing spine. O-arm (Medtronic Inc, Louisville, KY, USA), which is a cone beam CT that provides 3D imaging is one of the latest intraoperative imaging platforms to allow real-time multidimensional surgical imaging. It has become increasingly popular in the spinal surgery during the last decade [12,13]. Advanced intraoperative CT-based imaging and navigation increase both cost and operation time. Therefore, their use needs to be justified by improved safety and outcome [9].

Our strategy has been to use free-hand pedicle screw insertion and to verify the screw positioning with intraoperative 2D imaging using C-arm, and more recently, 3D imaging using O-arm before the corrective maneuvers. In this study, we aimed to evaluate the complications, need for reoperations, and health-related quality of life (HRQOL) of 3D imaging (O-arm) as compared to 2D imaging (C-arm) in a consecutive series of AIS patients treated with posterior spinal fusion. We hypothesized that intraoperative 3D imaging would reduce pedicle screw-related complications compared to 2D fluoroscopy facilitating the identification of significant pedicle screw malposition.

## 2. Materials and Methods

This is a retrospective analysis of patients with AIS treated with PSI and evaluated with 2D or 3D intraoperative imaging. Patients were treated in an academic tertiary medical center between 2009–2021. The indication for surgery was a primary curve greater than a Cobb angle of 45°. Patients underwent a posterior spinal fusion with bilateral segmental pedicle screw instrumentation using a combination of vertebral column derotation techniques and selective translation, compression, as well as a distraction [5]. The inclusion criteria of the study were a diagnosed AIS, age between 10–21 years, and posterior spinal fusion with segmental PSI. Exclusion criteria were bleeding disorder, Chiari malformation or syringomyelia in magnetic resonance images, and need for anteroposterior approach or vertebral column resection. All procedures were performed by the same senior orthopedic spine surgeon and patients had a minimum of six months of follow-up (mean 2.0 years, range 0.5–3.2). The study received approval from the Ethics Committee of the Hospital District (ETMK 96/1801/2020).

Intraoperative 3D-imaging scan using the O-arm (Medtronic Inc., Louisville, KY, USA) became available at our institution in 2016. There were 198 consecutive adolescents, of which 101 (51%) were evaluated during 2009–2016 with intraoperative 2D imaging scan (designed as a 2D group) and 97 (49%) were evaluated during 2016–2021 with intraoperative 3D imaging scan (designed as a 3D group). Intraoperative spinal cord (motor evoked and somatosensory potentials) and lumbar nerve root monitoring (electromyography, EMG) were performed in every patient in both study groups. Prospectively collected institutional pediatric spine register was used to acquire data including clinical characteristics, radiographic parameters, health-related quality of life outcomes, and complications.

### 2.1. Intraoperative Imaging

Pedicle screws in the 2D group were intraoperatively evaluated using traditional methods including posteroanterior and lateral fluoroscopic imaging after all screws were placed. Non-harmonious screws or screws violating bony landmarks were identified. In the 3D group, O-arm spins of every screw were obtained at the end of screw placement. This required typically two different imaging sessions as the height of the one imaging area was limited to five or six levels. Settings included standard thoracic values for the lumbar and lower thoracic area and high-density values in the upper thoracic area.

### 2.2. Surgical Technique

All patients were treated with bilateral segmental pedicle screw instrumentation for adolescent idiopathic scoliosis. Pedicle screws were inserted using the free-hand technique [5,14] by the senior orthopedic spine surgeon or under his direct supervision. The screw placement was standardized including three pairs of polyaxial screws on top of the construct, uniplanar screws in the midthoracic spine, a single polyaxial screw in the concave apex, and polyaxial screws in the lumbar spine. Screw diameters varied from 4.5 mm to 6.5 mm.

The decision to replace a screw in the 2D group was based on an intraoperative neuromonitoring (IONM) event (decrease of 50% or more in the motor evoked potential) or a pedicle screw violating the bony landmarks on the 2D imaging. Intraoperative neuromonitoring was performed with 15-min interval and at standard procedure time points: before incision, exposure completed, pedicle screws implanted, correction completed, and wound closed. Fluoroscopic 2D imaging was performed after all screws were placed. In the 3D group, intraoperative neuromonitoring events and intraoperative 3D imaging were used for the decision to replace screws. Intraoperative 3D imaging scans were evaluated by the senior surgeon and misplaced screws of 2 mm or more were replaced during the index procedure.

### 2.3. Scoliosis Research Society-24 (SRS-24) Outcome Questionnaire

The SRS-24 is a disease-specific health-related quality of life (HRQOL) questionnaire used to assess the current state of the patient with AIS and the outcomes of scoliosis surgery. The questionnaire has seven domains: pain, general self-image, function from a back condition, the general level of activity, postoperative self-image, postoperative function, and satisfaction. Each domain score ranges from 1 to 5, with higher scores indicating better patient outcomes. The maximum raw score of SRS-24 is 120 points (corresponding mean maximal score of 5.0 points) [15].

### 2.4. Statistical Analyses

The normal distribution assumption of the data was verified visually with QQ-plot and with the Shapiro–Wilk test. Descriptive statistics were presented in absolute numbers and percentages or means with standard deviations (SDs) or ranges. Statistical comparisons between the groups were performed with the chi-squared test for categorical parameters and with an independent-samples t-test for continuous variables. All analyses were conducted in JMP^®^ for Macintosh, Version 16.1 (SAS Institute Inc., Cary, NC, USA, 1989–2021). *p* values < 0.05 were considered statistically significant.

## 3. Results

The mean age of the patients was 15.5 ± 2.1 years, and the mean Cobb angle of the main curve was 52° ± 8.2° at the time of the surgical procedure, with a remaining curve of 13° ± 4.7° at the final follow-up. There were 1931 pedicle screws in the 3D group and 2048 in the 2D group. There were no significant differences in age, gender, radiographic correction of the deformity, or SRS-24 total score at six months or two-year follow-ups between the two groups (Table 1).

### 3.1. Need for Screw Repositioning

In the 3D group, 44 (45%) of the patients had at least one pedicle screw repositioned in the index operation, as compared with 13 (13%) in the 2D group (*p* < 0.001). A total of 70 screws (3.6% of all screws, average 0.73 per patient, range 0–4) were replaced in the index procedure in the 3D group, as compared with 13 (0.63%, average 0.13 per patient, range 0–2, *p* < 0.001) in the 2D group. In the 3D group, 43 of the replaced screws breached the medial wall (61%, mean breach 3.2 mm, range 2–6.5 mm), 17 breached the lateral wall of the pedicle (24%, mean breach 3.6 mm, range 2–6), and 10 the anterior cortex (15%, mean breach 2.7 mm, range 1.1–5 mm) of the vertebrae. Four patients in the 3D group and one in the 2D group had a screw replaced due to a sagittal breach. After screw repositioning, screw placement was verified using 2D imaging in both groups. One patient had transient IONM change (correction maneuver) requiring no replacement of screw in the 3D group. Nine patients in the 2D group had a screw removed intraoperatively due to an IONM signal change.

### 3.2. Complications

Complications are listed in Table 2. There were no new neurological deficits related to screw misplacement in the 3D group compared to six (5.9%) in the 2D group (*p* = 0.015). The more detailed description of new neurologic deficits and outcomes are represented in Table 3. Five (4.9%) patients in the 2D group required reoperation due to pedicle screw complications compared to no reoperations in the 3D group (*p* = 0.009).

Postoperative CT scans were performed to the six patients with new neurological deficits in the 2D group. Of the five patients requiring screw removal in the 2D group, two were performed during the index procedure and three in a separate procedure. Four patients in the 2D group had a new motor deficit ranging from muscle weakness to paresis. Three of these patients underwent reoperation within 48 h after the index surgery. One patient with fractured L4 pedicle causing chronic pain and atrophy of quadriceps muscles was treated conservatively with frequent monitoring. Two patients in the 2D group developed postural headache due to delayed dural lesions and required reoperation at approximately three months after the initial surgery. None of the patients had deep surgical wound site infection. Three patients in the 2D group, and none of the patients in the 3D group had superficial surgical site infection (*p* = 0.043). None of the patients requiring reoperation had wound-related complications or infections.

Four of the patients with neurological complications in the 2D group had a full recovery during the follow-up. Two patients developed a permanent neurological deficit. One patient had persistent mild spinal cord deficit (brisk reflexes) and right-sided motor and sensory impairment on the trunk. Another patient had a fractured pedicle and developed denervation of quadriceps muscle in electromyography, which lead to quadriceps muscle atrophy and chronic pain at follow-up.

## 4. Discussion

We present findings on the value of advanced intraoperative 3D imaging after free hand pedicle screw placement in adolescent idiopathic scoliosis. Advanced intraoperative 3D imaging reduced pedicle screw-related complications and reoperations in adolescents undergoing pedicle screw instrumentation for idiopathic scoliosis as compared to intraoperative 2D fluoroscopic imaging.

Free-hand pedicle screw instrumentation based on anatomic bony landmarks is fast and effective and carries a relatively low risk for neural deficits. The accuracy of navigation is limited due to movement between the reference and the instrumented level in the mobile growing spine often resulting in lateral deviation of the screw tip [9]. Patients with pedicle screws placed with CT-guided navigation have a lower rate of severely malpositioned screws and unplanned returns to the operating room than patients with pedicle screws placed with freehand/fluoroscopic technique [16,17,18]. A study on spinal navigation showed a significant shortening of the operation time as more experience was gained, thus demonstrating a positive effect on the learning curve [19].

There exist previous studies reporting the value of navigation-based pedicle screw instrumentation in this spinal condition [11,20]. CT scan is the standard imaging method for evaluating pedicle screw breaches. Plain radiographs have poor reliability in detecting pedicle screw breaches. A cadaveric study on intraoperative 3D-imaging scans reported high accuracy on significant breaches, but lower accuracy on breaches under 2 mm due to artifact signals [10]. However, minor breaches are considered probable safe zone [21]. Kobayashi et al. reported similar radiation exposure in intraoperative and preoperative CT scans suggesting that low dose protocol might help reduce the overall intraoperative dosage when compared to fluoroscopy [22]. Intraoperative 3D imaging reduces the radiation dosage for the surgical team, as it allows more efficient radiation protection.

Insertion of pedicle screws in pediatric patients is challenging due to the small osseous elements of the growing spine. The rotational deformity further complicates the estimation of the screw positioning [23]. However, the elastic nature of immature pedicles allows enlargement of the bony walls, and therefore, screws up to 115% of the pedicle diameter can be implanted [7]. Studies with computer tomography (CT) have reported a wide range of misplaced screws. The largest study of 2020 screws reported a 20.3% perforation rate [24]. Most of the moderate breaches of the pedicle are asymptomatic and overall, PSI has a relatively low complication rate in pediatric scoliosis surgery [6,24]. However, delayed complications caused by previously unrecognized breaches have been reported [25,26,27]. The surgical operations on the spine have gradual learning curves and there are many intraoperative difficulties and complications which can be encountered. Advanced intraoperative 3D imaging provides a useful aid for less experienced surgeons, as it provides immediate feedback and extra information during the surgery.

A recent study based on prospective adolescent idiopathic scoliosis database found an overall 0.4% incidence of return to the operating room due to screw malposition between years 2003 and 2017. Return to OR for screw malposition changed from 2003 to 2017 (1–0.2%) [28]. Moderate evidence shows CT guidance has lower point estimates of breach rates than free-hand methods at 7.9% compared with 9.7–17.1%. Screw-related complication rates are conflicting at 0% in CT navigation compared with 0–1.7% in 13 low- and moderate-quality studies [29,30].

Long-term outcomes of asymptomatic misplaced pedicle screws have not been extensively reported. In the literature, delayed postural headaches have been reported three months after the index procedure [26,31]. The intraoperative dural lesion is usually noticed during the screw placement due to cerebrospinal fluid leakage. The reports of delayed cerebrospinal fluid leak may indicate perforation of the dura over time by moderately breached screws. We hypothesize that most pedicle screw complications could be prevented by intraoperative evaluation of the screw positioning using 3D imaging after screw placement.

In our study, the screw revision rate was 3.9%. This is in accordance with literature. Larson et al. [11] reported high accuracy of pedicle screws inserted with intraoperative CT based navigation with a screw revision rate of 3.9%, but even after navigated screw insertion, a control CT scan was performed. In their study, the screw revision rate was significantly higher in pediatric than in adult patients. In the study of Sembrano et al. 602 pedicle screws were evaluated with an intraoperative CT scan and 2.8% were intraoperatively revised [32]. The HRQOL was assessed using the SRS-24 questionnaire. Despite the neurological complications in the 2D group, there were no statistically significant differences in 6 months or 2-year follows ups.

Further research is needed on the use and accuracy of navigation in pediatric spine. Evaluation of free-hand pedicle screws using advanced 3D imaging provides an intermediate method to maintain both effectiveness and safety of pedicle screw instrumentation in adolescents undergoing surgery for idiopathic scoliosis.

### Limitations

The current study was a retrospective evaluation of the value of advanced intraoperative 3D imaging on the safety of freehand pedicle screw placement in consecutive series of adolescents undergoing posterior spinal fusion for idiopathic scoliosis. Ideally, such evaluation should be performed in a randomized clinical trial. It is possible that the reduced number in the reoperations and complications in the 3D group might reflect the learning curve of the surgical team. However, the use of intraoperative 3D-imaging revealed significant number of pedicle screw breaches which were corrected during the index surgery. As CT scans were not routinely available for the 2D group, no comparison could be made in the accuracy of the screw placement between the groups. The study design was a single center, single surgeon series including 198 consecutive patients from 2009 to 2021 and not all patients had a minimum two-year follow-up. Clinical (complications), radiographic, and health-related quality of life data was based on a prospective spine register. The current study did not include cost-effectiveness analysis which should be investigated in the future.

## 5. Conclusions

Intraoperative 3D-imaging reduced pedicle screw related complications and reoperations in adolescents undergoing pedicle screw instrumentation for idiopathic scoliosis as compared with traditional intraoperative fluoroscopic 2D imaging. However, the improved clinical outcome was not reflected by better health-related quality of life at the final follow-up. Furthermore, 3D imaging provides immediate educational feedback for less experienced surgeons.

## Figures and Tables

**Table 1 children-09-01129-t001:** Clinical characteristics of the study groups.

Characteristics	3D Group (*n* = 97)	2D Group (*n* = 101)	*p*-Value
Number of pedicle screws	1931	2048	
Age at surgery, years	15.5 (10.5–21.9)	15.5 (10.7–22.5)	0.867
Female gender, *n* (%)	67 (69%)	75 (74%)	0.418
Follow-up time, years	1.8 (0.50–2.6)	2.1 (0.54–3.2)	<0.001
Major curve, degrees			
Preoperative	52 (45–83)	53 (45–84)	0.138
Postoperative	13 (2–28)	12 (2–24)	
Repositioning of at least one pedicle screw at the index procedure	44 (45%)	13 (13%)	<0.001
SRS-24 total score			
Preoperative	3.7 (2.4–4.6)	3.8 (2.5–4.4)	0.462
6 months follow-up	3.9 (2.4–4.6)	3.8 (2.7–4.7)	0.393
2-year follow-up	4.1 (2.5–4.7)	4.0 (2.9–4.6)	0.271

**Table 2 children-09-01129-t002:** Complications in the study groups.

Complication	3D Group (*n* = 97)	2D Group (*n* = 101)	*p* Value
Neurologic complications *	0	6	0.015
Motor deficit	0	4	0.048
Intraoperative monitoring change **	1	9	0.011
Cerebrospinal fluid leak	2	3	0.683
Intraoperative	2	1	0.534
Delayed	0	2	0.100
Surgical site infection			
Superficial	0	3	0.043
Deep	0	0	NA ***

* Includes new postoperative motor deficits as well as postural postoperative headache. ** Decrease of the motor evoked potential of 50% or more intraoperatively. *** Not applicable.

**Table 3 children-09-01129-t003:** New postoperative neurologic deficits and outcome.

Patient	Age at Surgery, Years	Follow-Up Time, Years	Levels	Neurologic Deficit	Actions Taken	Outcome
1	15.2	2.1	T4–L2	Isolated paresis of L5 level	Re-operation. T12 screw compressing cord, screw removed and a local decompression.	Full recovery at follow-up.
2	18.1	2.0	T2–L3	Post-operative muscle weakness in the left lower extremity.	Re-operation. T5 screw compressing cord and screw removed	Full recovery at follow-up.
3	19.2	2.0	T6–L3	Postural headache	Re-operation. Pedicle screw removed, saturation of the dura.	Full recovery at follow-up.
4	16.0	2.1	T5–L4	Motoric denervation in L3 and L4 myotomes according to ENMG	Frequent monitoring	Fracture healed and fragments disappeared in control CT. Mild quadriceps atrophy and pain after exercise at follow-up
5	16.7	2.0	T11–L3	Right sided muscle weakness in lower extremity	Re-operation. T11 pedicle screw removed. Decompression using enlarged posterior column osteotomy presented with no cause. Index operation stopped.	Mild spinal deficit at follow-up. Right side trunk muscle weakness and sensory deficit in lower extremity.
6	15.3	2.0	T6–L3	Postural headache	Re-operation. Pedicle screw removed, saturation of the dura	Full recovery at follow-up.

## Data Availability

Study data can be obtained from the corresponding author for a reasonable request.

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
