# Peer review of "Intraoperative 3D Imaging Reduces Pedicle Screw Related Complications and Reoperations in Adolescents Undergoing Posterior Spinal Fusion for Idiopathic Scoliosis: A Retrospective Study"

_children, 2022, doi:10.3390/children9081129_

Round 1
Reviewer 1 Report
Dear Authors,
It is my pleasure to review your study but I have a few doubts.
General information:
- this is retrospective study, this information should be contain in the title
Introduction:
-the introduction should contain newer references, especially that 3-D imaging is modern diagnostics,
-the article is also about Posterior Spinal Fusion , there is little information about it, it should be supplemented,
M&M:
-Intraoperative spinal cord (motor evoked and somatosensory potentials) and lumbar nerve root monitoring (electromyography, EMG) were performed in both study groups - I understand that every patient has had this procedure?
-Inclusion and exclusion criteria should be added,
- in 2.1. Intraoperative imaging - Pedicle screws in the 2D group were intraoperatively evaluated using traditional methods including posteroanterior and lateral fluoroscopic imaging after all screws were placed.
but in 2.2. we have:
The decision to replace a screw in the 2D group was based on an intraoperative neuromonitoring event (decrease of 50% or more in the motor evoked potential) or a pedicle 109 screw violating the bony landmarks on the fluoroscopic imaging at the end of screw placement.
I am not sure, when was 2-D imaging? After inserting all of screws? or after each screw? Then we have different radiation exposure.
-You described in 2.3 SRS24, ok. However, there is no information about this analysis in the title, abstracts and the aim of the study ?
Renault:
-table 1 shows all the information, it needs to be sorted / divided / organized. It should be corrected,
-please explain the abbreviation IONM,
-in table 3, please provide units for age and follow-up time,
Discussion:
-the analysis of the SRS24 questionnaire was presented, no information about it in the discussion. It needs to be sorted out.
References:
-newer references should be added.
Reviewer 2 Report
Thank you for the opportunity to review this article.
The study compares the accuracy of the free-hand and CT-guided technique in pedicle screw placement in patients with AIS.
The article is well written in all its sections, the methodology is correct, the design is appropriate although weak. Indeed, there are many limitations of retrospective studies with cohorts of patients separated by time criteria, which the Authors have adequately reported.
Honestly, this article, taken alone, adds very little to the literature on the topic. However, it may be useful to expand the available evidence, also for future data pooling in the literature. I therefore suggest to accept the article for publication, after minor revisions. Therefore, I would ask the authors for a couple of changes:
1) the discussion of screw accuracy is detailed, it would be useful in my opinion to report more data regarding neurological complications reported in similar studies;
2) in general, there are in my opinion more studies than the ones cited regarding the comparison of free-hand versus CT-guided in screw placement; it would be useful to report their conclusions in the discussion section.
Thank you.
Round 2
Reviewer 1 Report
Dear Authors,
Currently, the article looks much better. A correction has been made.
I have no objections. I think it can be published in Children (ISSN 2227-9067).